

# Global radiative effects of solid fuel cookstove aerosol emissions
Yaoxian Huang[1], Nadine Unger[2], Trude Storelvmo[3], Kandice Harper[1], Yiqi Zheng[3], and Chris
Heyes[4]
[1]School of Forestry and Environmental Studies, Yale University, New Haven, CT 06511, USA
[2]College of Engineering, Mathematics and Physical Sciences, University of Exeter, Exeter, EX4
4QE, UK
[3]Department of Geology and Geophysics, Yale University, New Haven, CT 06511, USA
[4]International Institute for Applied Systems Analysis, Laxenburg, Austria
*Correspondence to:* Y. Huang (yaoxian.huang@yale.edu)
**Abstract**. We apply the NCAR CAM5-Chem global aerosol-climate model to quantify the net
global radiative effects of black and organic carbon aerosols from global and Indian solid fuel
cookstove emissions for the year 2010. Our updated assessment accounts for the direct radiative
effects, changes to cloud albedo and lifetime (aerosol indirect effect, AIE), impacts on clouds via
the vertical temperature profile (semi-direct effect, SDE), and changes in the surface albedo of
snow and ice (surface albedo effect). In addition, we provide the first estimate of household solid
fuel black carbon emission effects on ice clouds. Anthropogenic emissions are from the IIASA
GAINS ECLIPSE V5a inventory. A global dataset of black carbon (BC) and organic aerosol (OA)
measurements from surface sites and aerosol optical depth (AOD) from AERONET is used to
evaluate the model skill. Compared with observations, the model successfully reproduces the
spatial patterns of atmospheric BC and OA concentrations, and agrees with measurements to
within a factor of 2. Globally, the simulated AOD agrees well with observations, with normalized
mean bias close to zero. However, the model tends to underestimate AOD over India and China
by ~ 19% but overestimate it over Africa by ~ 25%. Without BC serving as ice nuclei (IN), global
and Indian solid fuel cookstove aerosol emissions have a net cooling impact on global climate of
-141 ± 4 mW m$^{-2}$ and -12 ± 4 mW m$^{-2}$, respectively. The net radiative impacts are dominated by
the AIE and SDE mechanisms, which originate from enhanced cloud condensation nuclei
concentrations for the formation of liquid and mixed-phase clouds, and a suppression of convective
transport of water vapor from the lower troposphere to the upper troposphere/lower stratosphere
that in turn leads to reduced ice cloud formation. When BC is allowed to behave as a source of IN,



the net global climate impacts of the global and Indian solid fuel cookstove emissions range from
-260 to +135 mW m$^{-2}$ and -33 to +24 mW m$^{-2}$, with globally averaged values -51 ± 210 and 0.3 ±
29 mW m$^{-2}$ respectively. The uncertainty range is calculated from sensitivity simulations that alter
the maximum freezing efficiency of BC across a plausible range: 0.01, 0.05 and 0.1. BC-ice cloud
interactions lead to substantial increases in high cloud (< 500 hPa) fractions. Thus, the net sign of
the impacts of carbonaceous aerosols from solid fuel cookstoves on global climate (warming or
cooling) remains ambiguous until improved constraints on BC interactions with mixed-phase and
ice clouds are available.
**1. Introduction**
Worldwide 2-3 billion people rely on solid fuels for the majority of their energy needs (Legros et
al., 2009). This household biomass combustion includes burning wood fuel, agricultural residues
and dung for cooking, heating and lighting. Emissions from household solid fuel combustion
include greenhouse gases (carbon dioxide and methane), black carbon (BC), organic carbon (OC),
and other trace gases (e.g., nitrogen oxides). Globally, BC from household solid fuel emissions
accounts for approximately 25% of the total anthropogenic BC emissions (Bond et al., 2013).
Among different types of cookstoves, advanced charcoal stoves show lowest BC emission factors,
followed by simple charcoal, advanced biomass, rocket and simple wood stoves, respectively
(Garland et al., 2017). In India, residential biofuel combustion represents the dominant energy
sector and accounts for over 50% of the total source of BC and OC emissions (Klimont et al.,
2009). BC-rich household solid fuel emission plays an important role in affecting regional air
quality (Archer-Nicholls et al., 2016; Carter et al., 2016; Liu et al., 2016) and influencing global
climate change (Bauer et al., 2010; Butt et al., 2016; Venkataraman, 2005). The human health
consequences of solid fuel combustion are substantial (Archer-Nicholls et al., 2016; Ezzati and
Kammen, 2002; Lelieveld et al., 2015). Nearly 9% of the global burden of disease is attributable
to exposure to household air pollution from solid fuels, equivalent to 2.9 million premature deaths
and 86 million disability adjusted life years (DALYs) annually (GBD 2015 Risk Factors
Collaborators, 2016). Half of the world's population is exposed to indoor air pollution, mainly
attributable to solid fuel usage for household cooking and heating (Bonjour et al., 2013; Smith et
al., 2014).



Carbonaceous aerosols from solid fuel combustion interact with the Earth's radiation budget
directly by absorbing and scattering solar radiation (direct radiative effect, DRE) and indirectly by
changing cloud albedo and lifetime (aerosol indirect effect, AIE), modifying the vertical
temperature profile (semi-direct effect, SDE), and changing the surface albedo over snow and ice
(surface albedo effect, SAE) (Boucher et al., 2013; Chung, 2005; Chylek and Wong, 1995; Ghan,
2013; Ghan et al., 2012; Myhre et al., 2013). Carbonaceous aerosols affect cloud albedo and
lifetimes (the AIE) by acting as cloud condensation nuclei (CCN) or ice nuclei (IN), thus
modifying cloud properties and changing the top-of-atmosphere (TOA) radiative fluxes
(Lohmann, 2002; Lohmann et al., 2000; Penner et al., 1992; Pierce et al., 2007; Spracklen et al.,
2011b). The net climatic effect of carbonaceous aerosols from household solid fuel combustion is
not well constrained and even the sign is uncertain (Bond et al., 2013). Bauer et al. (2010)
estimated that the aerosol net global climate impact of residential biofuel carbonaceous aerosol
emissions is -130 mW m$^{-2}$. Kodros et al. (2015) have estimated that net DRE of solid fuel aerosol
emissions ranges from -20 to +60 mWm$^{-2}$, AIE from -20 to +10 mWm$^{-2}$, with uncertainties due to
assumptions of the aerosol emission masses, size distribution, aerosol optical properties and
mixing states. Butt et al. (2016) reported that the DRE and AIE of the residential emission sector
(including coal) ranged from -66 to +21 mW m$^{-2}$, and from -52 to -16 mW m$^{-2}$, respectively.
However, neither of the latter two studies consider the aerosol cloud-lifetime effect (second
indirect effect), SDE and SAE. From the perspective of policy-relevant country-level assessment
of cookstove burning on global climate, Lacey and Henze (2015) revealed that solid fuel cookstove
aerosol emissions resulted in global air surface temperature changes ranging from 0.28 K cooling
to 0.16 K warming, concluding that emissions from China, India and Ethiopia contributed the most
to the global surface temperature changes (Lacey et al., 2017).
None of the previous assessments have included BC-ice cloud interactions that can exert a large
influence on the atmospheric radiation balance. A recent study by Kulkarni et al. (2016) showed
that BC could act as IN, which was also shown by past lab and field findings (Cozic et al., 2008;
DeMott et al., 1999; Koehler et al., 2009). With BC as IN, Penner et al. (2009) estimated that the
total radiative forcing of anthropogenic and biomass BC emissions was -300 to -400 mW m$^{-2}$, with
IN parameterizations following Liu and Penner (2005) and Kärcher et al. (2006). Gettelman et al.
(2012) further concluded that AIE from BC emissions was -60 mW m$^{-2}$, with IN parameterization
following Barahona and Nenes (2009). Hence, a re-assessment of the global climate change



impacts of carbonaceous aerosol emissions from the solid fuel cookstove sector that newly
incorporates BC as IN is urgently needed.
Here, we employ a global aerosol-climate model to quantify the impacts of solid fuel cookstove
carbonaceous aerosol emissions globally and from India on global climate change. Sect. 2 presents
the Methods including the evaluation measurement data sets for BC, OA and aerosol optical depth
(AOD), the model description and experimental design. Sect. 3 details the results of the model
evaluation and the impacts of the global and Indian solid fuel cookstove emissions on the
atmospheric radiation budget and global climate. Discussion and summary are provided in Sect.

98    4.

## 99    2. Methods


### 101    2.1 BC and OC evaluation measurement database

Ground-based BC observations are from IMPROVE (the Interagency Monitoring of PROtected
Visual Environment, http://vista.cira.colostate.edu/Improve/) for the year 2010 over North
America (Malm et al., 1994), EMEP (the European Monitoring and Evaluation Programme,
http://ebas.nilu.no) for 2009-2013 over Europe, and sporadic measurement campaigns for China
and India. Elemental carbon (EC) concentrations are measured using Thermal Optical Reflectance
(TOR) (Chow et al., 1993, 2004; EMEP/MSC-W et al., 2014). Our measurement database
comprises a total of 152 sites from IMPROVE, 28 sites from EMEP, 35 sites for China, and 41
sites for India. The number of urban sites includes 8 from IMPROVE, 5 from EMEP, 17 for China,
and 23 for India.
A global network of aerosol mass spectrometer (AMS) surface measurements for organic aerosol
(OA) for 2000-2008 are used to compare with model simulations (Spracklen et al., 2011a; Zhang
et al., 2007; Zheng et al., 2015). The AMS technique measures hydrocarbon-like OA (HOA),
oxygenated OA (OOA) and total OA (HOA + OOA). HOA is a surrogate for primary OA (POA)
emitted directly from fossil fuel and biomass burning, while OOA is a surrogate for secondary OA
(SOA). In this study, we compare monthly mean total OA with model simulated total OA (POA +
SOA). The majority of the AMS measurements in the surface concentration database were made
prior to 2005.



Ground-based    AOD    observations    from    AERONET    (AErosol    RObtic    NETwork,
https://aeronet.gsfc.nasa.gov) during 1993-2016 are applied to examine model skill  (Dubovikl and
King, 2000; Holben et al., 1998, 2001). A climatological AOD value averaged over 1993-2016 for
each site is used to compare with the model simulation. The AERONET version 2 level-2 product
is used in this study.
**2.2 NCAR CAM5-Chem global model description**
We apply the NCAR Community Atmosphere Model version 5.3 with chemistry (CAM5-Chem)
within the Community Earth System Model (CESM) version 1.2.2 (Emmons et al., 2010;
Lamarque et al., 2012; Tilmes et al., 2015). The oxidant-aerosol system is fully coupled in CAM5-
Chem. The horizontal resolution of CAM5-chem is 0.9° latitude by 1.25° longitude, with 56
vertical levels from surface up to about 40 km. In the standard CAM5-Chem, aerosol
microphysical processes are represented using a 3-mode scheme (MAM3; aitken, accumulation
and coarse modes). MAM3 simulates both mass and number concentrations of aerosols. Aerosol
size distributions in each mode are assumed to be lognormal (Liu et al., 2012). The model treats
the effects of aerosol acting as CCN in liquid-phase clouds (Ghan et al., 2012). The aerosol
components in MAM3 include BC, primary organic matter (POM), secondary organic aerosol
(SOA), sulfate, sea salt and dust, which are assumed to be internally mixed within each lognormal
mode. Mass yields of semi-volatile organic gas-phase species (SOAG) from emissions of isoprene,
monoterpenes, big alkanes and alkenes, as well as toluene are prescribed (Emmons et al., 2010;
Liu et al., 2012; Tilmes et al., 2015). The condensable SOAG reversibly and kinetically partitions
into the aerosol phase to form SOA in CAM5-Chem as described in Liu et al. (2012).
**2.3 Emissions**
Global anthropogenic emissions are from the IIASA (International Institute for Applied System
Analysis) Greenhouse Gas-Air Pollution Interactions and Synergies (GAINS) integrated
assessment model ECLIPSE V5a (Evaluating the Climate and Air Quality Impacts of Short-lived
Pollutants version 5a) for the year 2010 (Amann et al., 2011, 2013; Klimont et al., 2017; Stohl et
al., 2015). Species in ECLIPSE V5a include BC, POM, sulfur dioxide, nitrogen oxides, carbon
monoxide, volatile organic compounds, and ammonium, with their annual global budgets for the
year 2010 shown in Table 1. ECLIPSE V5a emissions available at 0.5° latitude by 0.5° longitude



spatial resolutions are re-gridded to the model spatial resolution. ECLIPSE V5a does not include
shipping or wildfire biomass burning emissions, which are instead obtained from the IPCC AR5
RCP8.5 scenario for the year 2010 (Riahi et al., 2011).

**2.4 Simulations**

Atmosphere-only simulations are performed in specified dynamics (SD) mode with offline
meteorological fields from the Goddard Earth Observing System model version 5 (GEOS-5). In
this SD mode configuration, the internally derived meteorological fields (e.g., horizontal wind
component, air temperature and latent heat flux) are nudged by 10% towards reanalysis fields from
GEOS-5 for every model time step. The nudging technique in CAM5-Chem has been evaluated to
quantify the aerosol indirect effect in order to reduce the influence of natural variability
(Kooperman et al., 2012). Sea surface temperature and sea ice in the model are prescribed from
the Climatological/Slab-Ocean Data Model (DOCN) and Climatological Ice Model (DICE)
respectively, with monthly-varying decadal mean averaged over 1981-2010.
We perform three sets of model simulations using the model configurations shown in Table 2. The
first set of simulations represents the control with anthropogenic emissions following ECLIPSE
V5a, as described above (hereafter referred to as BASE). The second set of simulations are
identical to the BASE simulation except the global solid fuel cookstove emissions for aerosols and
gas-phase aerosol and ozone precursors are set to zero (termed as GBLSF_OFF). The third set of
simulations is identical to BASE except the solid fuel cookstove emissions are set to zero over the
Indian sub-continent (termed as INDSF_OFF). We run all the above simulations for 6 years from
2005 to 2010, with the first year discarded as spin-up and the last five years averaged for output
analysis. The differences between BASE and GBLSF_OFF isolate the impacts of the global solid
fuel cookstove sector aerosol emissions, and the differences between BASE and INDSF_OFF
isolate the impacts of the Indian solid fuel cookstove sector aerosol emissions. Top-of-the-
atmosphere (TOA) aerosol shortwave (SW) and longwave (LW) radiative effects are calculated
using the Rapid Radiative Transfer Model for GCMs (RRTMG) that is coupled to CAM5-Chem
(Ghan, 2013; Ghan et al., 2012).

**3  Results**

### 3.1 Evaluation of surface BC and OA concentrations





Surface observation networks from IMPROVE, EMEP, and various campaigns in China and India
are employed to compare with model simulations, as shown in Figure 1. We diagnose the
normalized mean bias (NMB) for each source region, calculated as
$\quad \text{NMB} = \left( \frac{\sum_i (M_i - O_i)}{\sum_i O_i} \right) \times 100\%$ $\hspace{4cm}$ (1)
where M and O represent monthly mean model simulated and observational concentrations at site
$i$ respectively, and $\sum$ is the sum over all the sites within a source region.
In general, the model simulated surface BC concentrations agree with observations to within a
factor of 2, consistent with previous studies (Huang et al., 2013; Wang et al., 2011, 2014a, 2014b).
A total of 41 surface BC observational sites are used to evaluate the model simulation over India
(Fig. 1a). On average, the model underestimates surface BC concentrations by approximately 45%
and 34% over urban and rural sites respectively, with a total NMB -41% (Fig. 1a), which implies
a marked underestimation of the BC emissions in India. Previous modeling studies have also
reported large underestimates of BC surface concentrations over India against observations
(Gadhavi et al., 2015; He et al., 2014; Zhang et al., 2015). Part of the model/measurement
discrepancy is related to a sampling bias because the majority of the observations are located over
urban or heavily polluted regions. For China sites, the NMB value is -16% (Fig. 1b). Similar to
India, the model substantially underestimates the surface BC concentrations over urban sites with
a NMB of -30%. However, the model performs relatively well over rural areas, with a NMB close
to zero. For IMPROVE, the NMB values for rural and urban sites are -15% and -43%, respectively,
with a total NMB -28% (Fig. 1c). Over Europe, the model simulated surface BC concentrations
agree quite well with observations, with a NMB value of -8%, although two urban sites show
substantial model underestimation (Fig. 1d).
The 40 AMS surface OA measurements are grouped into three categories: East Asia (8 sites),
North America (17 sites) and Europe (15 sites) (Spracklen et al., 2011a; Zhang et al., 2007; Zheng
et al., 2015). Figure 2 shows the evaluation of simulated surface OA against observations. Over
East Asia, the model slightly underestimates observed OA, with a NMB of -8.5% (Fig. 2a). In
contrast, the simulated OA concentrations overestimate the measurements by over a factor of 2 in
North America, with a NMB value of 124% (Fig. 2b). For the European sites, we find a simulated



OA overestimation of measured concentrations by up to 0.9 μg m⁻³, corresponding to a NMB of
+32% (Fig. 2c).
**3.2 Evaluation of model AOD**
Figure 3 compares simulated AOD values against observations over nine regions across the globe,
including India, China, Rest of Asia (excluding China and India), Africa, South America, North
America, Europe, Australia and remote regions. Over India, the simulated annual mean AOD is
lower than observations by about 16% (Fig. 3a), with large bias sources mainly from the northern
India regions (e.g., New Delhi and Kanpur). This is consistent with Quennehen et al. (2016) who
also reported that model simulated AOD values were generally lower than satellite-derived AOD
over northern India, using the same emission inventory as our study. As discussed in Sect. 3.1,
model simulated surface BC concentrations over India are also underestimated (by up to 41%),
therefore, the low bias of model simulated AOD can be attributed, in part, to the underestimation
of Indian BC emissions from ECLIPSE V5a emission inventory (Stohl et al., 2015), although
global anthropogenic BC budgets in ECLIPSE V5a lie in the high end compared with previous
studies (Bond et al., 2004, 2013; Janssens-Maenhout et al., 2015). A similar pattern is found over
China (Fig. 3b) and the rest of Asia (Fig. 3c), with NMB values of -21% and -15% respectively.
Model simulated AOD values from several sites in West Asia (Fig. 3c) are higher than
observations, which is probably caused by the model overestimation of dust emissions (He and
Zhang, 2014). This directly leads to annual mean model simulated AOD values over Africa 25%
higher than observations because Saharan dust emissions dominate the AOD over North Africa
(Fig. 3d). For South America, the model generally agrees quite well with observations (Fig. 3e),
except for a few sites where model simulated AOD values are lower than observations by more
than a factor of 2. This is probably due to the model underestimation of biomass burning emissions
there (Reddington et al., 2016). AOD values over North America (Fig. 3f) and Europe (Fig. 3g)
are relatively lower (with values generally < 0.3), due to lower anthropogenic emissions. In these
two regions, modeled AOD agrees with observations within a factor of 2, with NMB values -20%
and -18% respectively. CAM5-Chem overestimates AOD over Australia (Fig. 3h) and remote sites
(Fig. 3i), with NMB values of +69% and +47%, respectively. Globally, model simulated AOD
agrees quite well with observations, with NMB values close to zero.
**3.3 Contribution of solid fuel cookstove sector emissions to atmospheric BC and POM**



### 3.3.1 BC

Annual BC emissions and budgets are reported in Table 3 based on the anthropogenic inventory from ECLIPSE V5a. Annual BC emissions from the global and Indian solid fuel cookstove emissions are 2.31 and 0.36 Tg yr$^{-1}$, accounting for 23.7% and 3.7% of the total BC emissions. For the control simulation, global annual mean BC burden and lifetime are 0.12 Tg and 4.5 days, respectively (Table 3), at the low end of the range estimated by AeroCom (Schulz et al., 2006; Textor et al., 2006).

Figure 4 shows the zonal mean BC concentrations from the control simulation (Fig. 4a), global (Fig. 4b) and Indian (Fig. 4c) solid fuel cookstove emissions respectively. For the control simulation, in general, the highest BC concentrations (by up to 0.40 μg m$^{-3}$) occur at the surface over the emission source regions in the mid-latitudes (e.g., China and India). In the tropics and mid-latitudes, zonal mean BC concentrations decrease with increasing altitude, due to wet removal and deposition, as found in Huang et al. (2013). A similar vertical distribution is observed for the impacts from global and Indian solid fuel cookstove emissions, although the magnitude is smaller, compared with the control simulation. Annual mean BC burdens from global and Indian solid fuel cookstove emissions account for about 24% and 5% of that in the control simulation (0.12 Tg).

### 3.3.2 POM

Global POM emissions are mainly from biomass burning (31 Tg yr$^{-1}$) and anthropogenic emissions (18.9 Tg yr$^{-1}$), with global and Indian solid fuel cookstove emissions accounting for, 21% and 3.4% respectively, of the total POM emissions (Table 3). In our control simulation, the annual mean POM burden is 0.66 Tg, and the global annual mean POM lifetime is 4.8 days (Table 3).

In Figure 5, we show the annual zonal mean POM concentrations for the control simulation (Fig. 5a) and for global (Fig. 5b) and Indian (Fig. 5c) solid fuel cookstove emissions. There are two maxima in the annual zonal mean POM concentrations near the surface. One is located in the tropics due to the large biomass burning emissions there, and the other is located over mid-latitude regions and originates mainly from anthropogenic emissions (Chung and Seinfeld, 2002; Huang et al., 2013). For POM concentrations from global solid fuel cookstove emissions, a single maximum is evident in the Northern Hemisphere (NH) subtropics at the surface (Fig. 5b). The





surface maximum for the Indian solid fuel cookstove emissions reaches a maximum in the NH
subtropics. The annual mean POM burdens from global and Indian solid fuel cookstove emissions
are 0.13 Tg and 0.027 Tg respectively.
**3.4 Impacts of solid fuel cookstove aerosol emissions on global climate change**
**3.4.1 Direct radiative effect (DRE)**
The DRE impacts of the global and Indian solid fuel cookstove emissions are shown in Figure 6.
For the global solid fuel cookstove sector, the globally averaged DRE from aerosol emissions is
$+70 \pm 3$ mW m$^{-2}$ without treating BC as IN, which is a warming effect. The positive DRE from
global solid fuel cookstove emissions shows large spatial variability, with the largest impacts
located over western Africa, followed by India and China (figure not shown). The contributions of
BC and POM to DRE are $+105 \pm 4$ (warming) and $-14 \pm 1$ (cooling) mW m$^{-2}$, respectively. In other
words, the warming effect of BC is partially offset by the cooling effect from POM. Additional
cooling effects may come from sulfate and SOA. CAM5-Chem assumes that BC is internally
mixed with other components in the accumulation mode and simulates enhanced absorption when
BC is coated by soluble aerosol components and water vapor (Ghan et al., 2012), which results in
larger estimates of the DRE from BC (Bond et al., 2013; Jacobson, 2001b).
The DRE from Indian solid fuel cookstove emissions also corresponds to a net warming effect
(Fig. 6), with a global annual mean value of $+11 \pm 1$ mW m$^{-2}$. Large impacts are found over
continental India, the Tibetan Plateau and southeastern China. On a global annual basis, DRE
values from BC and POM emissions from the Indian solid fuel cookstove sector are $+18 \pm 1$ and $-$
$3 \pm 0.2$ mW m$^{-2}$, respectively.
**3.4.2  Aerosol indirect, semi-direct and surface albedo effects: BC not active as IN**
Global annual mean AIE and SAE values from global and Indian solid fuel cookstove aerosol
emissions are shown in Figure 6. In our study, AIE includes the first (albedo) and second (lifetime)
indirect effects, as well as the semi-direct effect. Annually averaged AIE from the global solid fuel
cookstove sector is $-226 \pm 5$ mW m$^{-2}$ (Fig. 6), with annual mean shortwave (SW) AIE $-122 \pm 22$
mW m$^{-2}$ and longwave (LW) AIE $-104 \pm 17$ mW m$^{-2}$, without treating BC as IN. Both the annual
mean SW and LW AIE thus yield cooling effects. The cooling signals of SW AIE mainly occur





292 over the western coast of South America, west and east coasts of Africa, South China and Himalaya

293 regions (figure not shown). This is directly linked to the contribution of global solid fuel cookstove

294 aerosol emissions to CCN (Pierce et al., 2007), which increases the cloud droplet number

295 concentrations (CDNC) and cloud liquid water path (CLWP). Figure 7 shows the global vertically-

296 integrated distribution of CLWP from the contribution of global solid fuel cookstove aerosol

297 emissions. The higher CLWP is due to the enhanced lifetime of liquid and mixed-phase clouds,

298 which therefore reflect more solar radiation, leading to cooling effect. For the LW AIE, the largest

299 cooling effect is found over tropical regions, especially over southern India and the Indian Ocean.

300 In order to investigate the causes of the LW AIE cooling effect, we analyze the cloud fraction

301 change over a defined region (Latitude:0-20ºN; Longitude:60-90ºE) due to the effect from the

302 global solid fuel cookstove sector. As shown in Figure 8a, cloud fraction in the lower troposphere

303 increases. However, in the middle and upper troposphere cloud fraction decreases by up to 0.6%,

304 with the strongest decrease found at ~150 hPa. We further analyze the changes in shallow and deep

305 convective mass fluxes of moisture over the same domain. As shown in Figure 8b, moist shallow

306 convective mass flux generally shows increases in the lower troposphere, which means that solid

307 fuel cookstove aerosol emissions enhance the convective transport of water vapor within the

308 boundary layer. By contrast, the deep convective mass flux demonstrates decreases from surface

309 up to the middle troposphere (Fig. 8c). This indicates that solid fuel cookstove aerosol emissions

310 may stabilize the boundary layer and inhibit the transport of water vapor from the surface to the

311 upper troposphere/lower stratosphere, which leads to decreases in ice cloud formation, thus

312 reducing cloud cover in the upper troposphere and lower stratosphere (UTLS) region at around

313 200 hPa (Fig. 8a) and a LW AIE cooling effect.

314 The global annual mean AIE from Indian solid fuel cookstove aerosol emissions accounts for

315 approximately 10% (-22 ± 3 mW m$^{-2}$) relative to the value of AIE from the global solid fuel

316 cookstove sector (Fig. 6), with globally averaged SW and LW AIE values of -3 ± 11 and -19 ± 11

317 mW m$^{-2}$ respectively.

318 Global annual mean SAE values from global and Indian solid fuel cookstove sector are relatively

319 small: +15 ± 3 and -2 ± 3 mW m$^{-2}$, respectively (Fig. 6). The warming effect is mainly due to the

320 deposition of BC on the surface of snow and sea ice (Flanner et al., 2007; Ghan, 2013; Ghan et al.,

321 2012).



### 3.4.3   Total radiative effect: BC not active as IN

The net total radiative effect of global and Indian solid fuel cookstove aerosol emissions are both cooling, with the global annual mean estimated to be -141 ± 4 and -12 ± 4 mW m$^{-2}$ respectively (Fig. 6). This suggests that if we remove solid fuel cookstove aerosol emissions, it will result in warming and thus slightly increased global surface air temperature. That being said, this is likely to be quite sensitive to model representation of aerosol mixing state (Fierce et al., 2017).

### 3.4.4   Total radiative effect: BC active as IN

In default CAM5-Chem, BC is not treated as IN (Liu et al., 2012; Tilmes et al., 2015). However, several lab and field studies have shown that BC particles could act as IN (Cozic et al., 2008; DeMott et al., 1999; Koehler et al., 2009; Kulkarni et al., 2016), as discussed in Section 1. Therefore, we conduct sensitivity studies in our model simulations by treating BC as an effective IN, with the ice nucleation scheme by Barahona and Nenes (2008, 2009). We run three additional model simulations, with model configurations identical to those in Table 2, except for the treatment of BC particles as effective IN. In addition, for each model simulation, we alter the plausible maximum freezing efficiency (MFE) of BC as 0.01, 0.05 and 0.1, from which the uncertainty ranges of the climatic impacts from global and Indian solid fuel cookstove aerosol emissions with BC as IN are quantified.

For the radiative effect of global solid fuel cookstove emissions with BC as IN, global annual mean DRE is 99 ± 12 mW m$^{-2}$, ranging from +85 to +107 mW m$^{-2}$, which is 21-53% higher than the DRE values from the default scheme (Fig. 6). Intriguingly, large globally averaged negative SW AIE (-1.33 ± 0.63 W m$^{-2}$) and positive LW AIE (+1.17 ± 0.44 W m$^{-2}$) for global solid fuel cookstove aerosol emissions are found, with annual mean values for the SW AIE ranging from -1.80 to -0.62 W m$^{-2}$ and from +0.66 to +1.44 W m$^{-2}$ for the LW AIE. This results in a rather uncertain net AIE, with a global annual mean AIE of -163 ± 216 mW m$^{-2}$ (Fig. 6). The reason for the large global annual average negative SW AIE and positive LW AIE is a substantial increase in high cloud (< 500 hPa) fractions when BC acts as an efficient IN. For instance, with MFE = 0.1, large increases (by up to 9%) in high cloud fractions from global solid fuel cookstove aerosol emissions are found over subtropical regions, especially over the southern Atlantic Ocean (Fig. 9). With BC particles active as IN, ice particle sizes become smaller, leading to a slower settling



velocity for ice particles and thus an increase in the lifetime of ice clouds. Increases in high clouds not only reflect more solar radiation back to space, but also trap more LW radiation within the troposphere. For SAE, the global annual mean value is $+14 \pm 8$ mW m$^{-2}$ (Fig. 6). As a result, the net total radiative effect of global solid fuel cookstove aerosol emissions ranges from -260 to + 135 mW m$^{-2}$, with a global annual mean of $-51 \pm 210$ mW m$^{-2}$ (Fig. 6). Again, the source of the large uncertainty of the total radiative effect is due to the choice of MFE values. With MFE = 0.01, the global mean LW AIE ($+660$ mW m$^{-2}$) outweighs SW AIE ($-620$ mW m$^{-2}$), and therefore results in a net warming effect. For other MFE values (0.05 and 0.1), the absolute global annual mean SW AIE values are always higher than the LW AIE, leading to a net negative (i.e., cooling) total radiative effect.

For the Indian solid fuel cookstove sector, the global annual mean net total radiative effect is 0.3 $\pm 29$ mW m$^{-2}$, with an AIE of $-18 \pm 37$ and a SAE of $+1 \pm 8$ mW m$^{-2}$, respectively.

## 4    Discussion and Summary

In this study, we employ the atmospheric component of a global 3-D climate model CESM v1.2.2, CAM5.3-Chem, to investigate the impacts of solid fuel cookstove emissions on global climate change. We update the default anthropogenic emission inventory using IIASA ECLIPSE V5a for the year 2010. We focus our analysis on the radiative effects of global and Indian solid fuel cookstove aerosol emissions. Model performance is evaluated against a global dataset of BC and OA measurements from surface sites and AOD from AERONET. Compared with observations, the model successfully reproduces the spatial patterns of atmospheric BC and OA concentrations, and generally agrees with measurements to within a factor of 2. Globally, the simulated AOD agrees quite well with observations, with NMB values close to zero. Nevertheless, the model tends to underestimate AOD values over source regions (except for Africa) and overestimate AOD over remote regions. The underestimates of AOD over India and China indicate that anthropogenic emissions of carbonaceous aerosols and sulfate precursors in ECLIPSE V5a are underestimated because carbonaceous aerosols and sulfate account for over 60% of the AOD over these two countries (Lu et al., 2011; Streets et al., 2009), which may introduce uncertainties for our climate estimates. In the control simulation, the global annual mean BC burden and lifetime are 0.12 Tg and 4.5 days. For POM, the burden and lifetime are 0.66 Tg and 4.8 days. Annual mean surface





BC (POM) concentrations over Northern India, East China and sub-Saharan Africa are 1.4, 0.74
and 0.11 µg m$^{-3}$ (6.5, 3.8 and 0.5 µg m$^{-3}$), respectively. BC and POM burdens from global solid
fuel cookstove emissions are 0.026 and 0.12 Tg, while contributions from the Indian sector are
0.005 and 0.024 Tg, respectively.
In the default CESM simulations without treating BC as IN, globally averaged DRE values from
global and Indian solid fuel cookstove emissions are +70 ± 3 and +11 ± 1 mW m$^{-2}$, respectively.
The contributions of BC and POM from global solid fuel cookstove emissions to the DRE are
+105 ± 4 and -14 ± 1 mW m$^{-2}$. Global annual mean SW and LW AIE values from global solid fuel
cookstove emissions are -122 ± 22 and -104 ± 17 mW m$^{-2}$, with contributions from India yielding
-3 ± 11 mW m$^{-2}$ for the SW AIE and -19 ± 11 mW m$^{-2}$ for the LW AIE, respectively. The cooling
effect of the SW AIE is associated with the increases of CCN and CDNC, whereas the negative
effects of LW AIE are caused by the suppression of convection that transports water vapor from
lower troposphere to upper troposphere/stratosphere, thus reducing ice cloud cover. The CAM5-
Chem also computes the SAE, with global and Indian solid fuel cookstove emissions contributing
+15 ± 3 and -2 ± 3 mW m$^{-2}$, respectively. As a result, the net total radiative effects of global and
Indian solid fuel cookstove emissions are -141 ± 4 and -12 ± 4 mW m$^{-2}$, respectively, both
producing a net cooling effect.
Sensitivity studies are carried out to examine the impacts of global and Indian solid fuel cookstove
emissions on climate by treating BC as an effective IN, with MFE as 0.01, 0.05 and 0.1,
respectively. For the climate impacts of global solid fuel cookstove emissions, global annual mean
DRE is +99 ± 12 mW m$^{-2}$, which is ~ 40% higher than the default model scheme in which BC
particles are not treated as IN (Fig. 6). This is driven by the increases of BC burden (due to
prolonged BC lifetimes) from global solid fuel cookstove emissions by up to 17% with BC as IN.
Because the BC absorption effect dominates the DRE, increases in BC burden enhance the
magnitude of annual mean DRE (Jacobson, 2001a). Compared with the default model scheme,
significant changes in globally averaged SW AIE are found, with a global annual mean of -1.33 ±
0.63 W m$^{-2}$, which is about an order of magnitude higher than that from the default scheme.
Moreover, in contrast to the cooling effect found in the default scheme, annual mean positive LW
AIE is simulated here (+1.17 ± 0.44 W m$^{-2}$). The above changes in cookstove emission induced
SW and LW AIE are caused by the substantial increases in high cloud (< 500 hPa) fractions with



BC particles acting as IN by up to 9% due to the effect of solid fuel cookstove emissions. Large
increases in high cloud fractions are found mainly over tropical regions, especially over southern
Africa. For the SAE, similar to the model default scheme, the global annual mean value is +14 ±
8 mW m$^{-2}$. Summing up the DRE, the AIE and the SAE, the net total radiative effect of global
solid fuel cookstove emissions is -51 ± 210 mW m$^{-2}$. For the Indian sector, the global mean total
radiative effect is 0.3 ± 29 mW m$^{-2}$, with a net AIE -18 ± 37 and a SAE +1 ± 8 mW m$^{-2}$,
respectively.
We compare our simulation results with previous studies as shown in Figure 10. The globally
averaged DRE in our control simulation is more than four times higher than that from the baseline
simulation of Kodros et al. (2015), which assumes homogeneous particle mixing state (Fig. 10).
Annual emissions of BC from global solid fuel cookstove sector in our study (2.3 Tg C yr$^{-1}$) is
approximately 44% higher than that from global biofuel emissions (1.6 Tg C yr$^{-1}$) in Kodros et al.
(2015), which, to some extent, leads to differences in annual mean DRE values together with
different optical calculations. The annual mean DRE value from another study by Butt et al. (2016)
differs from ours in magnitude and sign, and concluded that annually averaged DRE from
residential combustion sources was -5 mW m$^{-2}$ (Fig. 10). The negative effect of DRE in Butt et al.
(2016) is partially driven by the inclusion of $SO_2$ emissions (8.9 Tg $SO_2$ yr$^{-1}$) from commercial
coal combustion in the residential sector, leading to the cooling effect of sulfate and organic
aerosols outweighing the warming from BC. For AIE, our control simulation is 38 times higher
than that from Kodros et al. (2015) and over an order of magnitude higher than that from Butt et
al. (2016). Both Kodros et al. (2015) and Butt et al. (2016) used offline radiative models to
calculate AIE and only considered the first (albedo) aerosol indirect effect, which may partially
explain the AIE differences. As mentioned earlier, the AIE in our study includes aerosol first and
second indirect effects as well as the semi-direct effect. Lacey and Henze (2015) estimated that
the global surface air temperature changes due to solid wood fuel removal ranged from -0.28 K
(cooling) to +0.16 K (warming), with a central estimate of -0.06 K (cooling). This cooling estimate
is opposite to our study. However, we acknowledge that there are fundamental differences in
calculating the radiative effect between our study and Lacey and Henze (2015), which employed
absolute regional temperature potentials to quantify the climate responses.



Cookstove intervention programs have been implemented in developing countries, such as China,
India and some African countries, to improve air quality and human health and to mitigate climate
change (Anenberg et al., 2017; Aung et al., 2016; Carter et al., 2016). Our results suggest that
large-scale efforts to replace inefficient cookstoves in developing countries with advanced
technologies is not likely to reduce global warming through aerosol reductions, and may even lead
to increased global warming when aerosol-cloud interactions are taken into account. Therefore,
without improved constraints on BC interactions with clouds, especially mixed-phase and ice
clouds, the net sign of the impacts of carbonaceous aerosols from solid fuel cookstoves on global
climate (warming or cooling) remains ambiguous.

## Acknowledgements

This article was developed under Assistance Agreement No. R835421 awarded by the U.S.
Environmental Protection Agency to SEI. It has not been formally reviewed by EPA. The views
expressed in this document are solely those of the authors and do not necessarily reflect those of
the Agency. EPA does not endorse any products or commercial services mentioned in this
publication. N. Unger acknowledges support from the University of Exeter, UK. We are thankful
for helpful discussions with S. Tilmes and S. Ghan. This project was supported in part by the
facilities and staff of the Yale University High Performance Computing Center.

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



**Table 1. Annual budget for various species for the BASE, GBLSF_OFF and INDSF_OFF**
**simulations for the year 2010.**

| Specie | ECLIPSE V5a (BASE)[a] | GBLSF_OFF[a] | INDSF_OFF[a] |
|--------|------------------------|--------------|--------------|
| BC | 7.23 | 4.92 | 6.87 |
| POM | 18.9 | 8.53 | 17.2 |
| SO$_2$ | 98.5 | 97.1 | 98.37 |
| NO$_x$ | 120.5 | 118 | 119.8 |
| VOC | 81.1 | 52.4 | 76.6 |
| CO | 548 | 358 | 516 |
| NH$_3$ | 54.9 | 54.6 | 54.87 |

[a]Units are Tg specie/yr.














**Table 2. Model experiments setup.**

| Experiments | Anthropogenic emission scenario |
| --- | --- |
| BASE | ECLIPSE V5a |
| GBLSF_OFF | ECLIPSE V5a excluding global solid fuel cookstove emissions |
| INDSF_OFF | ECLIPSE V5a excluding Indian solid fuel cookstove emissions |



















**Table 3. Global budgets, burden and lifetime of BC and POM from model control**
**simulations.**

| Specie | BC | POM |
|---|---|---|
| Sources (Tg specie/yr) | 9.73 | 49.9 |
| fossil fuel and biofuel emissions | 7.23 | 18.9 |
| biomass burning emissions | 2.5 | 31 |
| Sinks (Tg specie/yr) | 9.72 | 49.8 |
| Dry Deposition | 1.8 | 8.14 |
| Wet Deposition | 7.92 | 41.7 |
| Burden (Tg) | 0.12 | 0.66 |
| Lifetime (days) | 4.5 | 4.8 |
















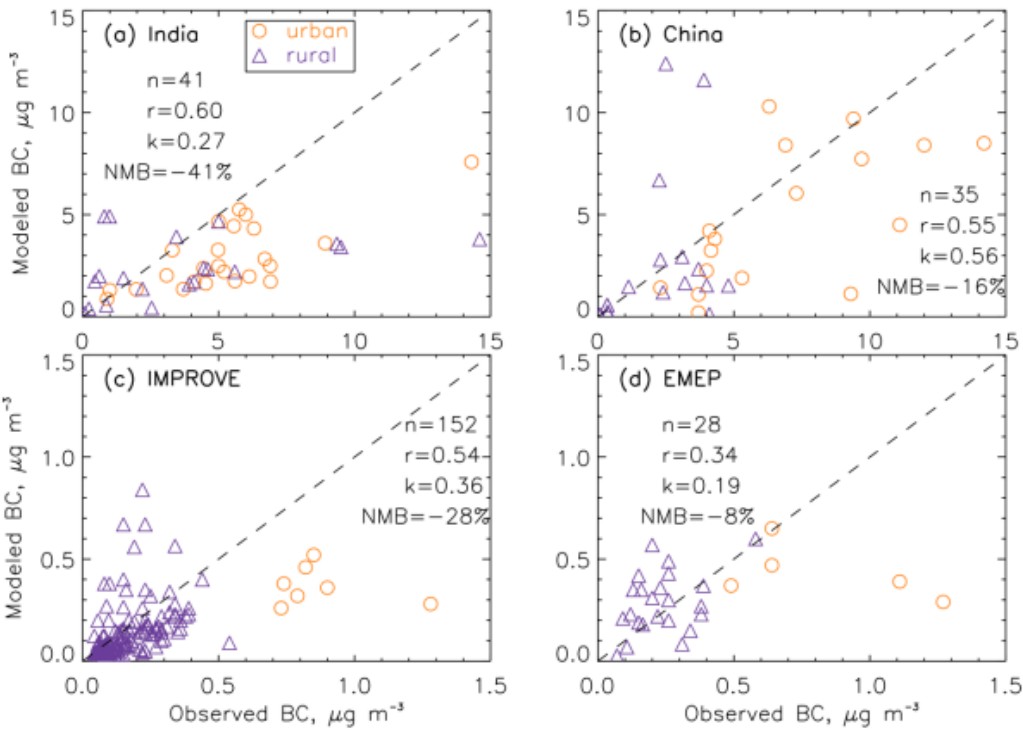


**Figure 1.** Comparisons of observational and model simulated annual mean surface BC concentrations from (a) India, (b) China, (3) IMPROVE, and (d) EMEP. Urban and rural sites are shown in orange circles and blue triangles for each region. For each panel, the total number of observational sites (n), model-to-observation regression slopes (k), correlation coefficient (r) and NMB values are included. The dashed line in each panel represents the 1:1 ratio.









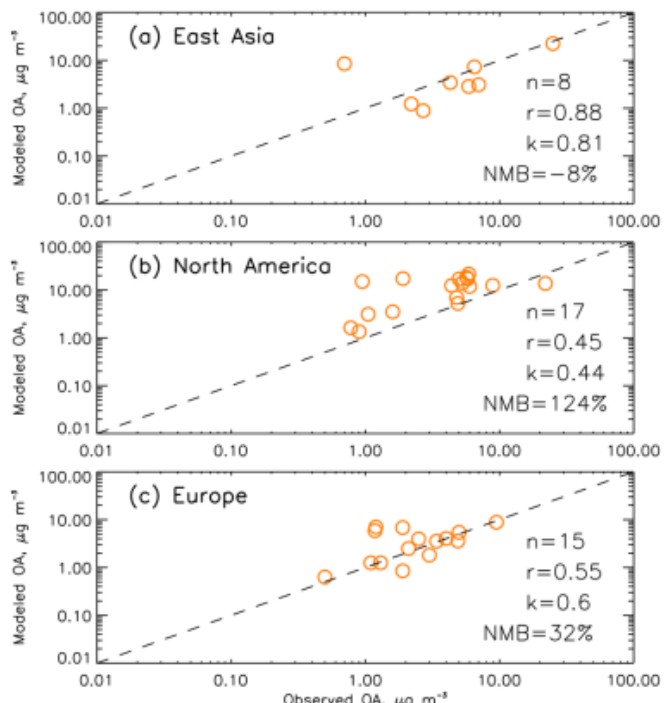


**Figure 2.** Comparisons of observational and model simulated surface OA concentrations from (a) East Asia, (b) North America, and (3) Europe. For each panel, the total number of observational sites (n), model-to-observation regression slopes (k), correlation coefficient (r) and NMB values are included. The dashed line in each panel represents the 1:1 ratio.

820

821





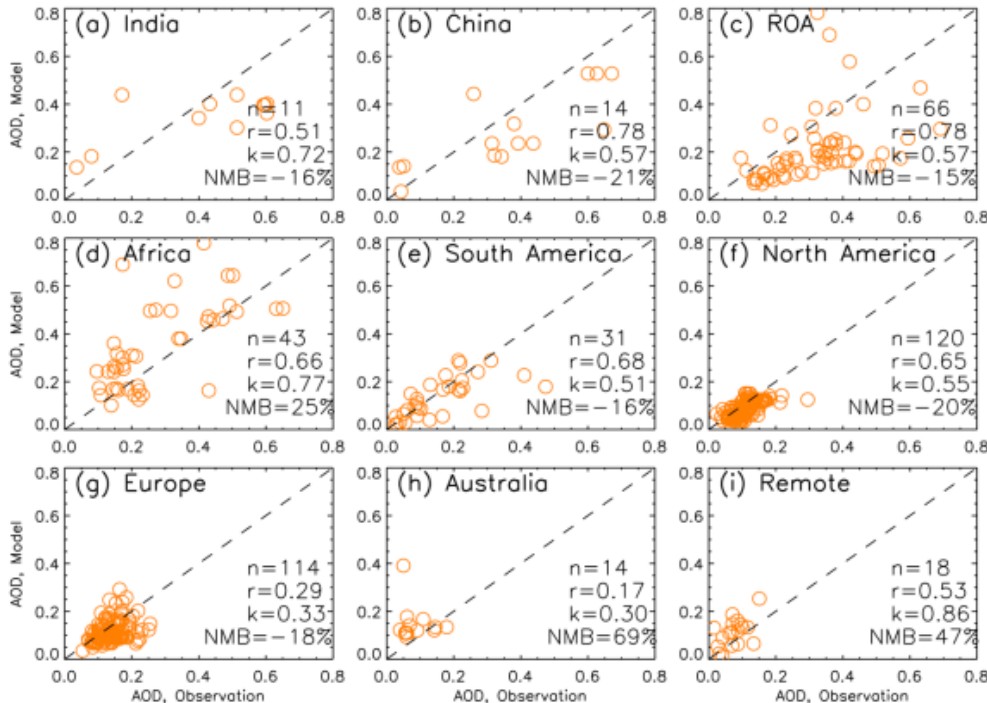

822

**Figure 3.** Scatter plots of AOD between model simulation and observations over (a) India, (b) China, (c) Rest of Asia (ROA), excluding China and India, (d) Africa, (e) South America, (f) North America, (g) Europe, (h) Australia and (i) Remote. For each panel, the total number of observational sites (n), model-to-observation regression slopes (k), correlation coefficient (r) and NMB are included.




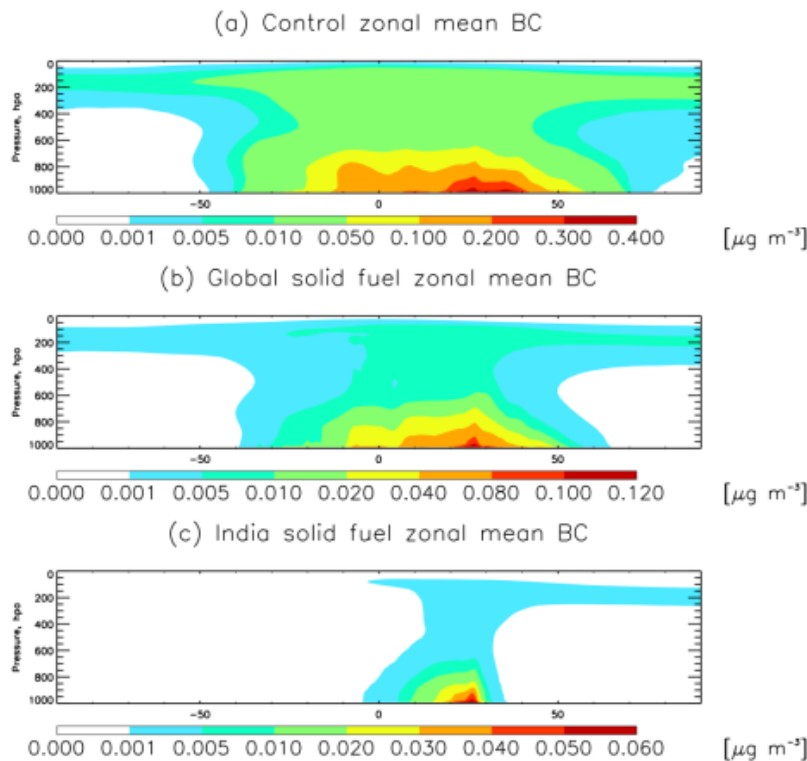


**Figure 4.** Annual zonal mean BC concentrations from (a) the BASE simulation, (b) the global and
(c) India solid fuel cookstove emissions.





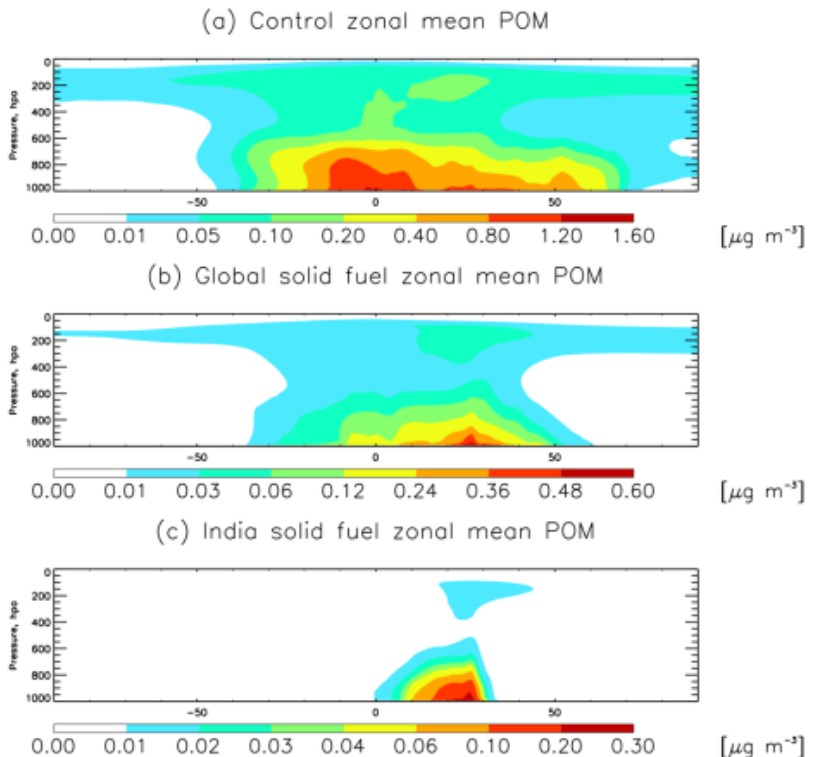


**Figure 5.** Same as Fig. 4 but for POM.





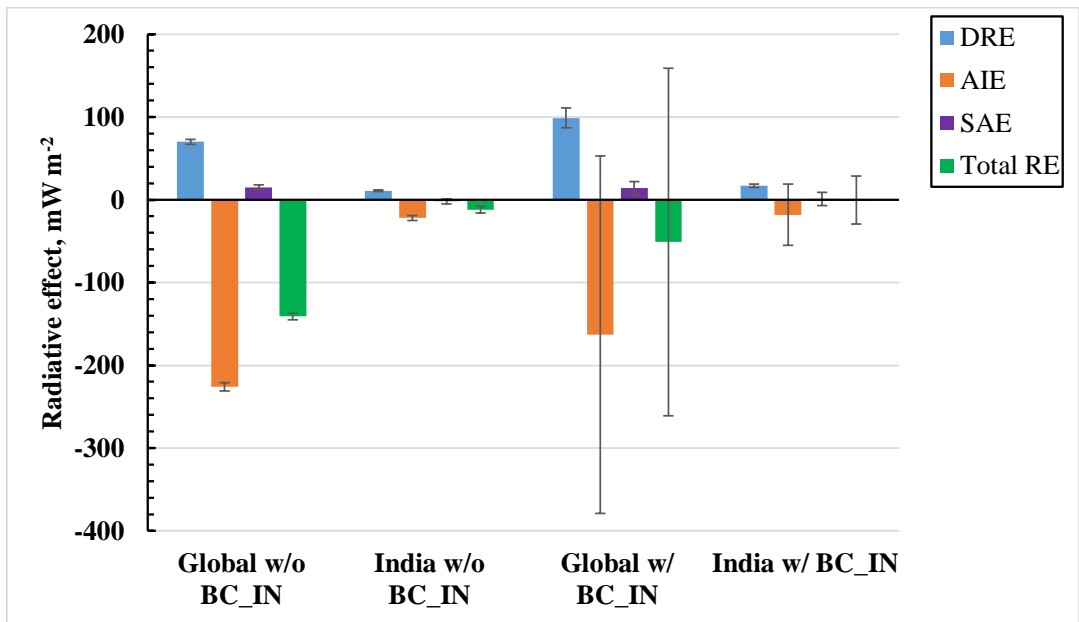


**Figure 6**. Radiative effect (RE) for global and Indian solid fuel cookstove aerosol emissions with BC not serving as IN (w/o BC_IN) and BC as IN (BC_IN), with DRE (blue), AIE (orange), SAE (purple) and total RE (green). Error bars represent one standard deviation for each RE. For BC as IN, standard deviations of RE are solely based on the choices of maximum freezing efficiency of BC as 0.01, 0.05 and 0.1 respectively.








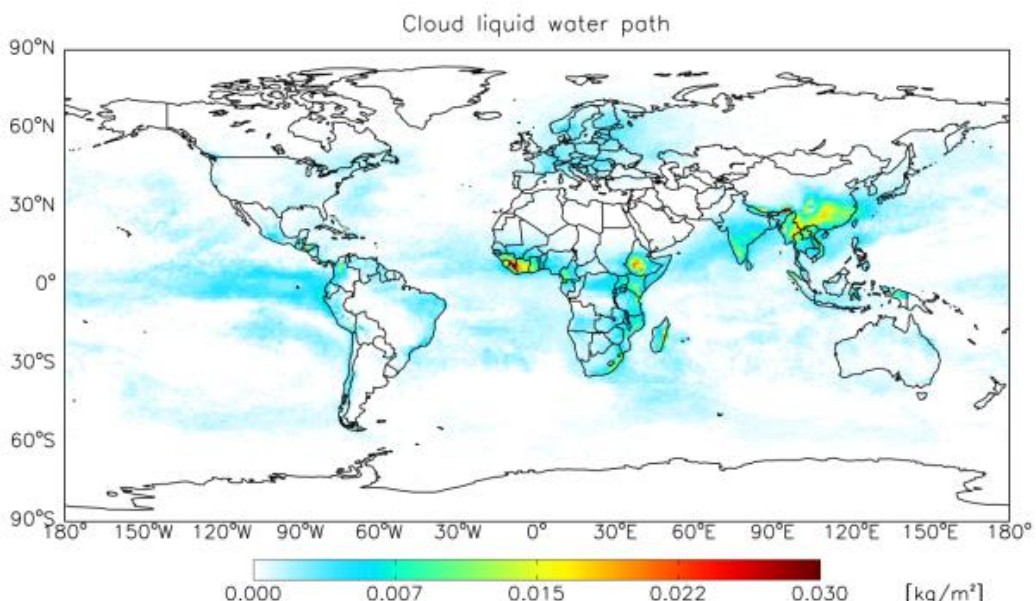


**Figure 7.** Global vertically-integrated cloud liquid water path from the global solid fuel cookstove

emissions.







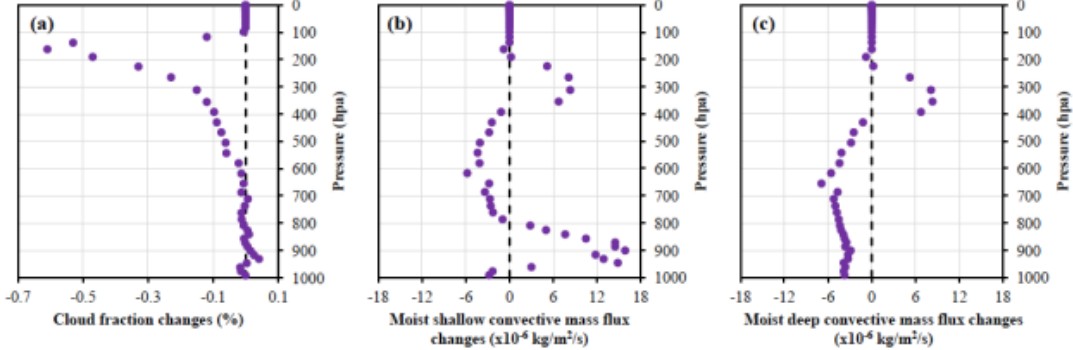


**Figure 8.** Changes in vertical cloud fractions (a), shallow (b) and deep (c) convective mass flux within the India and Indian Ocean domain from global solid fuel cookstove emissions.








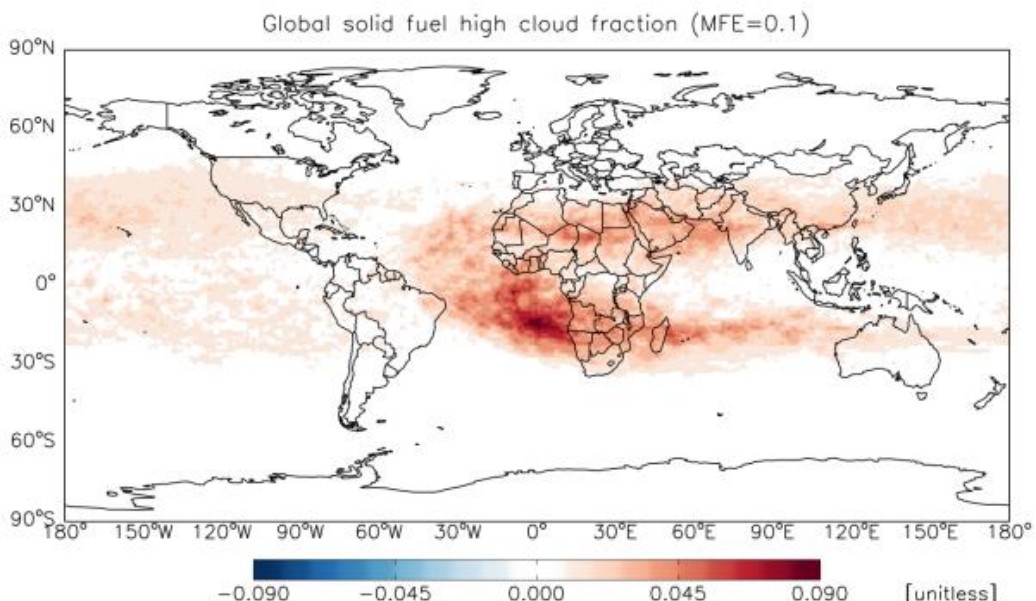


**Figure 9.** Global distribution of high cloud fraction due to solid fuel cookstove aerosol emissions
with BC as IN and MFE=0.1.












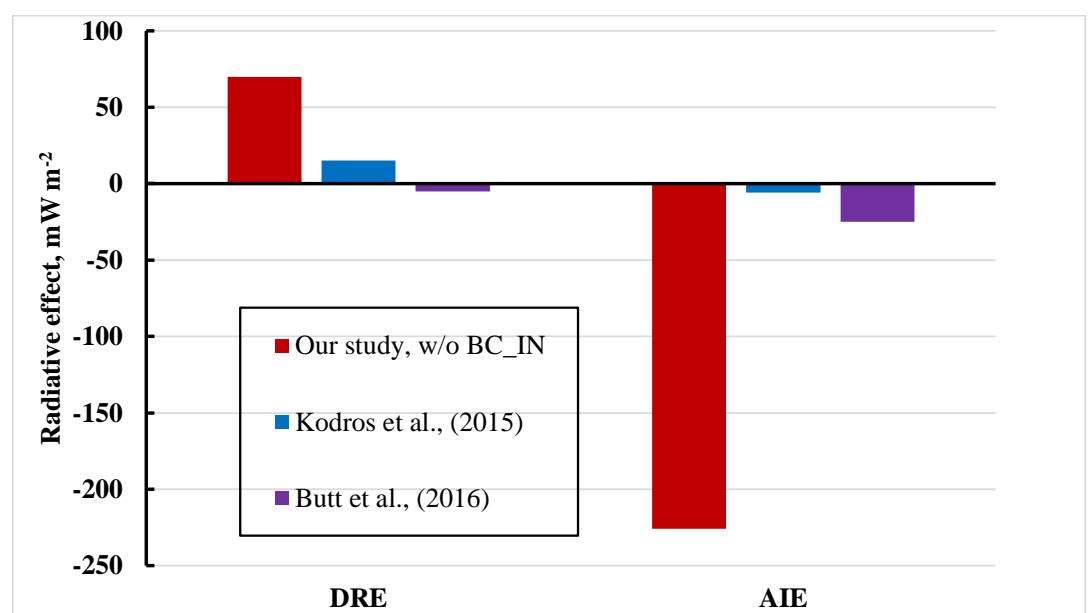



**Figure 10.** Comparisons of DRE (left) and AIE (right) radiative effects from global solid fuel
cookstove emissions in our control simulation with Kodros et al. (2015) and Butt et al. (2016).
