# Peer review of "Global radiative effects of solid fuel cookstove aerosol emissions"

_Atmospheric Chemistry and Physics, 2017_

## Referee Comment (RC1) · Anonymous Referee #2 · 4 Feb 2018

The manuscript by Huang et al. presents a modeling study of the radiative effects of solid fuel cookstoves, both globally and specifically in India. There is a lot of scientific and policy interest in this topic, given the potential climate co-benefts of cookstove intervention programs and the uncertainty associated with quantifying this, with several recent papers making a range of estimates that differ in sign. The contribution of this article is a welcome addition to the field, focusing particularly on details of aerosol cloud interactions, and considering the effect of BC ice nucleation, which has not been considered in such studies previously. Overall the manuscript is thus appropriate in scope for ACP. It is also generally clear, well organized, and easy to ready. I only have a few comments that are detailed below; there is some ambiguity regarding how the authors are arriving at their uncertainty estimates, and some of the motivation for the

scope of their analysis (e.g., considering just India, or not considering the impact of co-emitted GHGs) could be strengthened. Addressing these would constitute minor revision.

Major Comments:

37: An important conclusion, which I mainly agree with in spirit. However, it might be stated a little bit softer for a few reasons. First, the uncertainty of up to a factor of two in estimating concentrations would seem to contribute to the overall uncertainty in the net radiate effects. Second, this study is of radiative effects, not the climate response, and the wording should reflect that. Third, it's the result of only a single model, which may not be as definitive as presented.

The measurements used for comparison come from very different time periods (2010 for IMPROVE, 2009-2013 for Europe, 2000-2008 for AMS data, and 1993-2016 for AERONET). How does this impact the evaluation of modeled concentrations and AOD, given that the model uses emissions from 2010 and that there have been large changes in emissions over this time period?

215 - 220: Some previous studies have suggested that the resolution of global scale models leads to a bias that makes it difficult to match AOD from AERONET in these regions. Could this partially explain the low bias?

Fig 6 and Section 3.4.1 (and everywhere these numbers are quoted in the text): Suddenly the results have errors associated with them (concentrations and AOD did not. . .). What is the meaning of the error estimates? Are these the standard deviations over the timeframe modeled? If so, that needs to be more clearly stated when presenting these numbers in the abstract and conclusion (that +/- is modeled temporal standard deviation). And then I wonder why similar deviations were not considered for discussion of concentrations or AOD. Further, temporal variability is very different than e.g. an estimate of uncertainty owing to sources of model error or approximations, such as the ranges provided for the RF of the simulations including BC IN that stem from uncer-

tainty in the MFE. These ranges can't be directly compared, and yet they're presented in e.g. the abstract without distinguishing their different meanings. At present the non BC-IN ranges come across as uncertainty estimates that seem much too small (I doubt the authors believe that the aerosol RF in any single model could be that accurate).

Introduction: I didn't get a good sense from the introduction why there is a particular interest in India as separate from the globe in this study (as opposed to China, or any other country with significant cookstove use). I'm not suggesting that the authors do more simulations for other regions, but if they wish to include the India-specific results it would make more sense to include a bit more rational for this emphasis.

439 - 447: I think it's worth recognizing that there are climate impacts from GHG emissions as well. So, considering not just the aerosol emissions, these may be large enough to make the net climate images of cookstove emissions positive (Lacey 2017). It is somewhat artificial to envision a scenario wherein only the carbonaceous aerosol cookstove emissions are effected by stove replacements.

Minor Comments:

12: Not clear – "updated" related to what? A particular previous study? Later it becomes evident what is meant (first to include BC as IN), but perhaps it could be worded differently here.

74: Clarify whether the Butt 2016 study included just aerosols or also GHGs in the DRE.

80: Similarly, Ethiopia is the 3rd largest in Lacey 2017, but that's including GHGs in 2050, which is a slightly more specific statement than as presented here.

128: CAM5-Chem not CAM5-chem

Fig 1: Sorry if I just missed it, but did the authors state how they are defining urban vs rural in their classification of measurement sites?

[Figure]

276-279: What is the BC mass absorption coefficient (MABS) at 550 nm in this model? See e.g. Koch ACP 2009

---

## Referee Comment (RC2) · Anonymous Referee #1 · 6 Feb 2018

The paper reports a comprehensive study of the climate and air quality impacts of residential solid fuel combustion. The paper reports the first assessment of the impacts of carbonaceous aerosol from residential solid fuel combustion through changes in ice nuclei as well as a comprehensive assessment of aerosol indirect effects through changes in liquid clouds.

The paper makes an important contribution to our understanding of residential solid fuel combustion on climate and is suitable for publication in ACP. Importantly the manuscript highlights that the overall climate impact of carbonaceous aerosol from residential solid fuel combustion is uncertain and that even the sign of the impact is still ambiguous. The paper certainly motivates further study of this important issue and highlights where some of the major uncertainties lie.

[Figure]

The manuscript is very well written and clear. I only have a few minor comments. I suggest publication in ACP after the following minor comments have been addressed.

Minor comments

Page 5. Have you stated the size of the emitted carbonaceous aerosol? This is important for the impacts of carbonaceous aerosol on cloud condensation nuclei and aerosol indirect effects.

Page 12. Most models do not simulate the impacts of BC on ice nuclei. I think it would be useful to provide a brief summary of how this was treated in the Methods section of the paper.

Page 15. As stated by the authors the aerosol indirect effect is considerably larger than other studies. Ward et al. (2012) also found a large aerosol indirect effect using the CAM model (although a different version) to study carbonaceous aerosol from fires. It might be useful to point this out in the text.

Fig. 4 and 5. Please clarify whether these are reported at ambient conditions or at standard temperature and pressure.

References Ward et al., Atmos. Chem. Phys., 12, 10857–10886, 2012
* * *

---

## Author Comment (AC1) · 19 Mar 2018

**Response to Reviewer #1**

We thank Reviewer #1 for their valuable and helpful comments. Our responses to the comments are provided below in bold font with the reviewer's comments in italicized font.

*The paper reports a comprehensive study of the climate and air quality impacts of residential solid fuel combustion. The paper reports the first assessment of the impacts of carbonaceous aerosol from residential solid fuel combustion through changes in ice nuclei as well as a comprehensive assessment of aerosol indirect effects through changes in liquid clouds.*

*The paper makes an important contribution to our understanding of residential solid fuel combustion on climate and is suitable for publication in ACP. Importantly the manuscript highlights that the overall climate impact of carbonaceous aerosol from residential solid fuel combustion is uncertain and that even the sign of the impact is still ambiguous. The paper certainly motivates further study of this important issue and highlights where some of the major uncertainties lie.*

*The manuscript is very well written and clear. I only have a few minor comments. I suggest publication in ACP after the following minor comments have been addressed.*

*Minor Comments:*

*Page 5. Have you stated the size of the emitted carbonaceous aerosol? This is important for the impacts of carbonaceous aerosol on cloud condensation nuclei and aerosol indirect effects.*

**Response: We have added the size range of the emitted carbonaceous aerosols in the text (Page 5, Lines 155-156): "Specifically, BC and POM from solid fuel cookstove emissions are treated in the accumulation mode, with size range of 0.058-0.27 µm (Liu et al., 2012)."**

*Page 12. Most models do not simulate the impacts of BC on ice nuclei. I think it would be useful to provide a brief summary of how this was treated in the Methods section of the paper.*

**Response: We agree with the reviewer. We have moved part of the contents from Section 3.4.4 to a new section in Methods as Section 2.5 and added the description of BC as IN following Barahona and Nenes (2008, 2009) in the text (Page 7, Lines 197-212):**

**2.5 Simulations: BC active as IN**

**"In default CAM5-Chem, BC is not treated as IN (Liu et al., 2012; Tilmes et al., 2015). IN concentrations from homogeneous nucleation are calculated as a function of vertical velocity (Liu et al., 2007). Several lab and field studies indicate that BC particles can act as IN (Cozic et al., 2008; DeMott et al., 1999; Koehler et al., 2009; Kulkarni et al., 2016). Therefore, we conduct additional simulations that treat BC as an effective IN applying the ice nucleation scheme of Barahona and Nenes (2008, 2009). The scheme estimates maximum supersaturation and ice crystal concentrations and considers competition between homogeneous and heterogeneous freezing. Homogeneous nucleation occurs in solution droplets formed on soluble aerosols (mainly sulfate), while heterogeneous nucleation occurs on IN, which here are a small subset of mineral dust and black carbon particles. The heterogeneous freezing of BC and dust is described as a generalized ice nucleation spectrum.**

**We perform three additional model simulations, with model configurations identical to those in Table 2, except for the treatment of BC particles as effective IN. In addition, for each model simulation, we alter the plausible maximum freezing efficiency (MFE) of BC as 0.01, 0.05 and 0.1 that provides an uncertainty range in the global climatic impact assessment."**

*Page 15. As stated by the authors the aerosol indirect effect is considerably larger than other studies. Ward et al. (2012) also found a large aerosol indirect effect using the CAM model (although a different version) to study carbonaceous aerosol from fires. It might be useful to point this out in the text.*

**Response: We have added (Page 16, Lines 489-490): "Consistent with our study, Ward et al. (2012) also found a large AIE (-1.74 to 1.00 W m$^{-2}$) for carbonaceous aerosols from fires using CESM CAM4-Chem."**

*Fig. 4 and 5. Please clarify whether these are reported at ambient conditions or at standard temperature and pressure.*

**Response: We have clarified the reporting units for BC and POM under standard temperature and pressure, which have been added in the caption of Fig. 4 as (Page 36, Lines 916-917) "Figure 4. Annual zonal mean BC concentrations from (a) the BASE simulation, (b) the global and (c) India solid fuel cookstove emissions. BC concentrations are calculated under standard temperature and pressure conditions (273 K, 1 atm)."**

---

## Author Comment (AC2) · 19 Mar 2018

**Response to Reviewer #2**

We thank Reviewer #2 for their valuable and helpful comments. Our responses to the comments are provided below in bold font with the reviewer's comments in italicized font.

*The manuscript by Huang et al. presents a modeling study of the radiative effects of solid fuel cookstoves, both globally and specifically in India. There is a lot of scientific and policy interest in this topic, given the potential climate co-benefits of cookstove intervention programs and the uncertainty associated with quantifying this, with several recent papers making a range of estimates that differ in sign. The contribution of this article is a welcome addition to the field, focusing particularly on details of aerosol cloud interactions, and considering the effect of BC ice nucleation, which has not been considered in such studies previously. Overall the manuscript is thus appropriate in scope for ACP. It is also generally clear, well organized, and easy to ready. I only have a few comments that are detailed below; there is some ambiguity regarding how the authors are arriving at their uncertainty estimates, and some of the motivation for the scope of their analysis (e.g., considering just India, or not considering the impact of co-emitted GHGs) could be strengthened. Addressing these would constitute minor revision.*

*Major Comments:*
*37: An important conclusion, which I mainly agree with in spirit. However, it might be stated a little bit softer for a few reasons. First, the uncertainty of up to a factor of two in estimating concentrations would seem to contribute to the overall uncertainty in the net radiate effects. Second, this study is of radiative effects, not the climate response, and the wording should reflect that. Third, it's the result of only a single model, which may not be as definitive as presented.*

**Response: We prefer to keep the original statement as is. First, the manuscript includes a comparative description of all previous results with several different global aerosol-climate model frameworks that supports the uncertain sign conclusion for net global radiative effect of cookstove aerosol emissions. Second, the statement clearly refers to carbonaceous aerosols only and highlights the need for improved constraints on aerosol-cloud interactions. Finally, our study quantifies the impacts of cookstove carbonaceous aerosol emissions on global average annual mean radiative effect because it is a linear predictor of**

global average surface air temperature response at equilibrium. Our community is still several years away from any quantitative robust mechanistic understanding of regional climate response to regional radiative effect of aerosols (e.g. Kasoar et al., ACP, 2016). It would be fairly pedagogical to convert the global radiative effect results to global average surface air temperature response e.g. Berntsen and Fuglestvedt, PNAS, 2008.

*The measurements used for comparison come from very different time periods (2010 for IMPROVE, 2009-2013 for Europe, 2000-2008 for AMS data, and 1993-2016 for AERONET). How does this impact the evaluation of modeled concentrations and AOD, given that the model uses emissions from 2010 and that there have been large changes in emissions over this time period?*

Response: The measurement data availability and model configuration has necessitated using climatological observations for BC, OA and AOD with some apparent mismatches between observational years and simulations years for OA and AOD. Model simulated BC concentrations were sampled in exact correspondence to the observed temporal period. The global aerosol-climate model framework is a chemistry-climate model with specified dynamics (CCM-SD), not a chemistry transport model (CTM). CTMs exist to compare with exact measurement periods. In our case, the model output reflects a present-day climatology rather than a specific CTM run year as such. Therefore, the comparison with available climatological measurements does allow us to validate and have insights into the large-scale aerosol system dynamics and behavior. We clarify in the text (Pages 14-15, Lines 429-436): "The simulations reflect a present-day climatology forced with recycled year 2010 anthropogenic emissions. Model simulated BC concentrations were sampled in exact correspondence to the observed temporal period. In some limited cases, OA and AOD are not exactly temporally consistent with the available aerosol measurement network climatologies applied in the evaluation. For regions where carbonaceous aerosol emissions have undergone substantial changes over short periods, the model-measurement comparison may therefore introduce additional uncertainty. However, we focus the evaluation on the large-scale regional aerosol system dynamics."

*215 - 220: Some previous studies have suggested that the resolution of global scale models leads to a bias that makes it difficult to match AOD from AERONET in these regions. Could this partially explain the low bias?*

**Response: We agree and have added (Page 9, Lines 257-259): "The model underestimate of AOD from AERONET in India may also be related to the fairly coarse global model resolution, as previously reported by Pan et al. (2015) and Zhang et al. (2015)."**

*Fig 6 and Section 3.4.1 (and everywhere these numbers are quoted in the text): Suddenly the results have errors associated with them (concentrations and AOD did not: : :). What is the meaning of the error estimates? Are these the standard deviations over the timeframe modeled? If so, that needs to be more clearly stated when presenting these numbers in the abstract and conclusion (that +/- is modeled temporal standard deviation). And then I wonder why similar deviations were not considered for discussion of concentrations or AOD. Further, temporal variability is very different than e.g. an estimate of uncertainty owing to sources of model error or approximations, such as the ranges provided for the RF of the simulations including BC IN that stem from uncertainty in the MFE. These ranges can't be directly compared, and yet they're presented in e.g. the abstract without distinguishing their different meanings. At present the non BC-IN ranges come across as uncertainty estimates that seem much too small (I doubt the authors believe that the aerosol RF in any single model could be that accurate).*

**Response: We include uncertainty estimates that are based on interannual internal climate variability (n=5 years). For consistency, we have added the uncertainty ranges based on interannual internal climate model variability to concentrations and burdens of BC and POM, and AOD where multiple years have been sampled for the comparison. For example, model BC concentrations were sampled in correspondence to the exact temporal measurement period, thus no range included for that case. This revision is reflected throughout the updated manuscript, in the abstract, Section 3.1, Section 3.2, Section 3.3 and Section 4 e.g. (Page 1, Lines 24-26): "However, the model tends to underestimate AOD over India and China by ~ 19 ± 4% but overestimate it over Africa by ~ 25 ± 11%**

(uncertainty range due to interannual internal climate model variability for n=5 run years).”

(Pages 8-9, Lines 239-244): “Figure 2 shows the evaluation of simulated surface OA against observations. Over East Asia, the model slightly underestimates observed OA, with a NMB of -8.5 ± 5% (Fig. 2a). In contrast, the simulated OA concentrations overestimate the measurements by over a factor of 2 in North America, with a NMB value of 124 ± 24% (Fig. 2b). For the European sites, we find a simulated OA overestimation of measured concentrations by up to 0.9 ± 0.7 µg m$^{-3}$, corresponding to a NMB of +32 ± 26% (Fig. 2c).”

(Page 9, Lines 248-250): “Over India, the simulated annual mean AOD is lower than observations by about 16 ± 3% (Fig. 3a), with large bias sources mainly from the northern India regions (e.g., New Delhi and Kanpur).”

(Page 9, Lines 259-261): “A similar pattern is found over China (Fig. 3b) and the rest of Asia (Fig. 3c), with NMB values of -21 ± 4% and -15 ± 6% respectively.”

(Page 9, Lines 263-265): “This directly leads to annual mean model simulated AOD values over Africa 25 ± 11% higher than observations because Saharan dust emissions dominate the AOD over North Africa (Fig. 3d).”

(Pages 9-10, Lines 270-274): “In these two regions, modeled AOD agrees with observations within a factor of 2, with NMB values -20 ± 4% and -18 ± 9% respectively. CAM5-Chem overestimates AOD over Australia (Fig. 3h) and remote sites (Fig. 3i), with NMB values of +69 ± 17% and +47 ± 12%, respectively. Globally, model simulated AOD agrees quite well with observations, with NMB values close to zero.”

(Page 10, Lines 279-282): “For the control simulation, global annual mean BC burden and lifetime are 0.12 ± 0.001 Tg and 4.5 ± 0.04 days, respectively (Table 3), at the low end of the range estimated by AeroCom (Schulz et al., 2006; Textor et al., 2006).”

(Page 10, Lines 290-292): “Annual mean BC burdens from global and Indian solid fuel cookstove emissions account for about 24.2 ± 0.7% and 5.0 ± 0.0% of that in the control simulation (0.12 ± 0.001 Tg).”

(Page 10, Lines 296-298): “In our control simulation, the annual mean POM burden is 0.66 ± 0.006 Tg, and the global annual mean POM lifetime is 4.8 ± 0.04 days (Table 3).”

**(Page 11, Lines 307-308):** "The annual mean POM burdens from global and Indian solid fuel cookstove emissions are 0.13 ± 0.004 Tg and 0.027 ± 0.002 Tg respectively."

**(Page 15, Lines 436-442):** "In the control simulation, the global annual mean BC burden and lifetime are 0.12 ± 0.001 Tg and 4.5 ± 0.04 days. For POM, the burden and lifetime are 0.66 ± 0.006 Tg and 4.8 ± 0.04 days. Annual mean surface BC (POM) concentrations over Northern India, East China and sub-Saharan Africa are 1.55 ± 0.076, 0.76 ± 0.028 and 0.11 ± 0.004 $\mu g\ m^{-3}$ (7.11 ± 0.32, 3.95 ± 0.12 and 0.48 ± 0.02 $\mu g\ m^{-3}$), respectively. BC and POM burdens from global solid fuel cookstove emissions are 0.029 ± 0.001 and 0.13 ± 0.004 Tg, while contributions from the Indian sector are 0.006 ± 0.000 and 0.027 ± 0.004 Tg, respectively."

We have clarified in the abstract and text to distinguish the 2 uncertainty range calculations used in the study (1) based on interannual internal climate model variability (2) based on BC maximum freezing efficiency range **(Page 2, Lines 38-39):**
"Here, the uncertainty range is based on sensitivity simulations that alter the maximum freezing efficiency of BC across a plausible range: 0.01, 0.05 and 0.1."
**(Page 7, Lines 209-212):** "We perform three additional model simulations, with model configurations identical to those in Table 2, except for the treatment of BC particles as effective IN. In addition, for each model simulation, we alter the plausible maximum freezing efficiency (MFE) of BC as 0.01, 0.05 and 0.1 that provides an uncertainty range in the global climatic impact assessment."

*Introduction: I didn't get a good sense from the introduction why there is a particular interest in India as separate from the globe in this study (as opposed to China, or any other country with significant cookstove use). I'm not suggesting that the authors do more simulations for other regions, but if they wish to include the India-specific results it would make more sense to include a bit more rational for this emphasis.*

**Response: We have expanded the India regional focus motivation statement (Page 2, Lines 53-61):**

**"India contains a large concentration of solid fuel-dependent households: approximately 160 million households use solid fuels for cooking (Venkataraman et al., 2010). In India, residential biofuel combustion represents the dominant energy sector and accounts for over 50% of the total source of BC and OC emissions (Klimont et al., 2009). India has a long history of unsuccessful stove intervention programs that have sometimes focused on health benefits (Hanbar and Karve, 2002; Kanagawa and Nakata, 2007; Kishore and Ramana, 2002). Despite years of interventions, the vast majority of Indian households still rely on traditional stoves (Legros et al., 2009). The possible scope for global climate co-benefits in future Indian cookstove intervention programs warrants further examination and analysis of this region."**

*439 - 447: I think it's worth recognizing that there are climate impacts from GHG emissions as well. So, considering not just the aerosol emissions, these may be large enough to make the net climate images of cookstove emissions positive (Lacey 2017). It is somewhat artificial to envision a scenario wherein only the carbonaceous aerosol cookstove emissions are effected by stove replacements.*

**Response: We have added (Page 17, Lines 508-510) "This study does not include the greenhouse gas emission effects from the solid fuel cookstove sector, which may indeed be large enough to imply a net warming global climate impact depending on time scale (Lacey et al., 2017)."**

*Minor Comments:*
*12: Not clear – "updated" related to what? A particular previous study? Later it becomes evident what is meant (first to include BC as IN), but perhaps it could be worded differently here.*
**Response: We have deleted "updated".**

*74: Clarify whether the Butt 2016 study included just aerosols or also GHGs in the DRE.*
**Response: We have clarified (Page 3, Lines 86-88): "Butt et al. (2016) reported that the net DRE and AIE of aerosols from the residential emission sector (including coal) ranged from -66 to +21 mW m$^{-2}$, and from -52 to -16 mW m$^{-2}$, respectively. Their study did not include greenhouse gases."**

*80: Similarly, Ethiopia is the 3rd largest in Lacey 2017, but that's including GHGs in 2050, which is a slightly more specific statement than as presented here.*

**Response: We have revised to (Pages 3-4 Lines 89-95) "From the perspective of policy-relevant country-level assessment of cookstove burning on global climate, Lacey and Henze (2015) revealed that solid fuel cookstove aerosol emissions resulted in global air surface temperature changes ranging from 0.28 K cooling to 0.16 K warming; Lacey et al. (2017) further concluded that emissions reductions, including both aerosols and greenhouse gases, from China, India and Ethiopia contributed the most to the global surface temperature changes by 2050."**

*128: CAM5-Chem not CAM5-chem*
**Response: Corrected.**

*Fig 1: Sorry if I just missed it, but did the authors state how they are defining urban vs rural in their classification of measurement sites?*
**Response: We have added the definition of the classification of urban and rural sites in the text as (Page 5 Lines 128-129) "Here we define urban (including semi-urban) sites as the geographic locations of the measured sites locating in a city, others as rural sites."**

*276-279: What is the BC mass absorption coefficient (MABS) at 550 nm in this model? See e.g. Koch ACP 2009*

**Response: BC mass absorption cross section coefficient at 550 nm in CAM5-Chem is 14.6 $m^2$ $g^{-1}$. We have updated this in the revised text as (Page 11 Lines 319-323) "CAM5-Chem assumes that BC is internally mixed with other components in the accumulation mode and simulates enhanced absorption (BC mass absorption cross section = 14.6 $m^2$ $g^{-1}$) when BC is coated by soluble aerosol components and water vapor (Ghan et al., 2012), which results in larger estimates of the DRE than for BC alone (Bond et al., 2013; Jacobson, 2001b)."**

---

## Author Response (AR2)

**Response to the editor**

We thank the editor for their valuable and helpful comments. Our responses to the comments are provided below in bold font with the editor's comments in italicized font.

*Comments to the Author:*

*This paper looks good to go, however there are a couple of technical points raised by reviewer 2 that I do not feel have been adequately addressed and these need to be fixed before publication:*

*1. I'm afraid that I have to agree with the reviewer concerning the terminology of the impacts. Using the strict IPCC definitions, this work only deals with radiative impacts and while these will certainly have knock-on effects on climate, the two terms should not be conflated and as such, this work should not be presented as an evaluation of climate impacts. I would insist that terms like "the net global climate impacts" in the abstract (and elsewhere, e.g. lines 82, 458) are rephrased to "the net global radiative impacts". I would also change the title of section 3.4 to something like "Impacts of solid fuel cookstove aerosol emissions on radiative transfer"*

**Response: We agree with the editor. We have revised the term of "the net global climate impacts" in the text as "the net global radiative impacts" throughout the manuscript, e.g. (Page 1 Lines 26-29) "Without BC serving as ice nuclei (IN), global and Indian solid fuel cookstove aerosol emissions have a net global cooling radiative effects of -141 $\pm$ 4 mW m$^{-2}$ and -12 $\pm$ 4 mW m$^{-2}$, respectively ($\pm$ represents modeled temporal standard deviations for n=5 run years)."**

**(Page 2 Lines 36-38) "When BC is allowed to behave as a source of IN, the net global radiative impacts of the global and Indian solid fuel cookstove emissions range from -275 to +154 mW m$^{-2}$ and -33 to +24 mW m$^{-2}$, with globally averaged values -59 $\pm$ 215 and 0.3 $\pm$ 29 mW m$^{-2}$ respectively."**

**(Page 3 Lines 82-84) "Bauer et al. (2010) estimated that the net global radiative impact of residential biofuel carbonaceous aerosol emissions is -130 mW m$^{-2}$."**

**(Pages 15-16 Lines 442-445) "For the radiative impacts of global solid fuel cookstove emissions, global annual mean DRE is +105 $\pm$ 13 mW m$^{-2}$, which is ~ 50% higher than the default model scheme in which BC particles are not treated as IN (Fig. 6)."**

**We have also revised the title of Section 3.4 as (Page 11 Line 305) "3.4 Impacts of solid fuel cookstove aerosol emissions on global radiation budget".**

*2. I thank the reviewers for addressing the point concerning uncertainty, but they failed to address one key technical query of the reviewer, which was to clarify whether the plus or minus uncertainty ranges represent standard deviations or some other measure. This is very important and should be stated, both in the main text and the abstract.*

**Response: We have added the description of uncertainty ranges in the abstract and main text as**

[revised manuscript text omitted]